# Anisotropic, Hydrogel Microparticles as pH-Responsive Drug Carriers for Oral Administration of 5-FU

**DOI:** 10.3390/pharmaceutics15051380

**Published:** 2023-04-30

**Authors:** Serena P. Teora, Elada Panavaité, Mingchen Sun, Bas Kiffen, Daniela A. Wilson

**Affiliations:** Department of Systems Chemistry, Institute for Molecules and Materials, Radboud University, Heyendaalseweg 135, 6525 Nijmegen, The Netherlands; serena.teora@ru.nl (S.P.T.); panavaite.elada@gmail.com (E.P.); mingchen.sun@ru.nl (M.S.); b.kiffen@student.utwente.nl (B.K.)

**Keywords:** microfluidics, microgels, pH responsiveness, 5-FU, drug loading, in vitro drug release, oral administration, cytotoxicity

## Abstract

**Simple Summary:**

pH-responsive hydrogel microparticles have great potential as drug delivery systems of anti-cancer drugs. Here, we use microfluidics for the generation of asymmetric microgels as carriers for oral administration of 5-fluorouracil (5-FU) in colorectal cancer. Due to their anisotropic shape, they show increased on-demand loading of the drug. The pH-responsiveness is ensured by the presence of alginate methacrylate within the gel network and represents the key factor for 5-FU release at the targeted location. Empty, asymmetric microgels do not show cytotoxicity even at high concentration, while upon treatment with 5-FU loaded microparticles, the viability of tumor cells notably decreases confirming the efficacy of drug release at certain pH.

**Abstract:**

In the last 20 years, the development of stimuli-responsive drug delivery systems (DDS) has received great attention. Hydrogel microparticles represent one of the candidates with the most potential. However, if the role of the cross-linking method, polymer composition, and concentration on their performance as DDS has been well-studied, still, a lot needs to be explained regarding the effect caused by the morphology. To investigate this, herein, we report the fabrication of PEGDA–ALMA-based microgels with spherical and asymmetric shapes for 5-fluorouracil (5-FU) on-demand loading and in vitro pH-triggered release. Due to anisotropic properties, the asymmetric particles showed an increased drug adsorption and higher pH responsiveness, which in turn led to a higher desorption efficacy at the target pH environment, making them an ideal candidate for oral administration of 5-FU in colorectal cancer. The cytotoxicity of empty spherical microgels was higher than the cytotoxicity of empty asymmetric microgels, suggesting that the gel network’s mechanical proprieties of anisotropic particles were a better three-dimensional environment for the vital functions of cells. Upon treatment with drug-loaded microgels, the HeLa cells’ viability was lower after incubation with asymmetric particles, confirming a minor release of 5-FU from spherical particles.

## 1. Introduction

The design of modern drug delivery systems (DDS) that efficiently maintain the bioactivity of drug molecules and release therapeutic substances in a predictable manner has been the topic of many studies. In order to increase the efficacy of therapeutics and reduce their toxicity and required dosage, the ideal DDS should allow for the control of the drug availability to cells and tissues over time and in space. The achievement of these requirements is possible with hydrogel microparticles, which represent one of the most innovative strategies in drug delivery [1]. They offer the advantages of microsized structures in combination with the properties of a hydrogel: microparticles are small cargo-compartments very versatile in shapes and sizes, while the high-water hydrogel content (>95%) provides biocompatibility [2] and physical similarity to tissues [3]. Additionally, microgels are mainly generated in an aqueous environment, preventing the risk of drug denaturation due to the presence of organic solvents [4]. The performance of hydrogel microparticles as DDS can be mainly affected by their shape [5], by the polymers involved in the formation of the gel network [6], or by both these factors. The mechanical properties of solid-like microgels are influenced by the cross-linking method [7], polymer concentrations [8], and molecular weight [9]. Moreover, hydrogels can be developed in such ways that they undergo structural or mechanical changes in response to environmental triggers [10]. In this regard, stimuli-responsive hydrogels have been attracting interest over the last decades [11,12] as they represent bioinspired networks with biomimetic and biofunctional properties [13,14]. Different stimuli, such as light [15], magnetic [16] and electric fields [17], temperature [18,19], or pH [20] can be used to cause a phase transition, alter the stiffness, and change the mesh. With an accurate design, it is possible to fabricate very dynamic DDS with tuneable loading and release profiles [21,22,23,24], by carefully architecting the gel network’s characteristics and particles’ shape [25]. Although several spherical microparticles have been reported in drug delivery [4,26,27,28], less attention has been paid to anisotropic particles. We believe that asymmetric morphologies possess enhanced properties in comparison to their spherical counterparts in terms of cytotoxicity and therapeutics loading and release [29]. With this in mind, herein, we present a novel, dynamic, and asymmetric microgel system for the loading and in vitro pH-triggered release of 5-fluorouracil (5-FU). 5-FU is a fluoropyrimidine antimetabolite drug widely used in breast, colon, and skin cancer that inhibits essential biosynthetic processes and the normal function of DNA and RNA by being incorporated into macromolecules [30,31]. For the treatment of various malignancies, 5-FU is administered under continuous infusion regimens because of its time-dependent effects [32,33]. However, recent studies proved that orally administrable 5-FU drugs could replace the continuous infusion chemotherapies without significant changes in efficacy or side effects. Furthermore, oral administration prevents several iatrogenic issues, and clinical studies confirmed that patients prefer oral administration rather than continuous infusion procedures [34,35]. We used a microfluidic device for the generation of either spherical or asymmetric hydrogel microparticles made of a composite aqueous main phase of poly(ethylene glycol) diacrylate (PEGDA) and alginate methacrylate (ALMA). ALMA is a pH-responsive polymer obtained by the chemical modification of alginate. Alginate is an unbranched polysaccharide consisting of 1→4 linked β-d-mannuronic acid (M) and its C-5 epimer α-l-guluronic acid (G) [36]. It is extracted from brown seaweed, and it is widely used in pharmaceutical and biomedical applications due to its nonanimal origin, low toxicity, biocompatibility, and biodegradability [37]. Several alginate-based particles with controllable drug encapsulation efficiencies and release profiles have been reported [38,39]. These particles preserve the bioactivity of small molecules and various drugs, including proteins [40,41] and cytokines [42,43]. Their hydrogel network is commonly formed via ionic cross-linking of alginate with divalent cations [38,44,45,46], such as Ca^2+^ and Ba^2+^. However, due to exchange reactions with the environment and the migration of divalent cations from the alginate matrix, these gels present limited long-term stability [47]. To circumvent this drawback, alginate is functionalized with reactive groups that can be covalently cross-linked [48]. We used methacrylate-functionalized alginate in combination with PEGDA to prepare microgels by photochemical polymerization. The molecular weight, degree of methacrylate functionalization, cross-link density, and concentration of ALMA can be tuned to obtain hydrogels with different mechanical properties [48,49,50]. In this work, in both spherical and asymmetric morphologies, the concentration of PEGDA was kept at 30% *w*/*w*, while for ALMA, it was either 0.75% *w*/*w* or 1.5% *w*/*w*. The resulting microparticles were used to investigate the influence of shape and different concentrations of pH-responsive polymer within the gel network on their performance as DDS for the oral administration of 5-FU and their cytotoxicity.

## 2. Materials and Methods

### 2.1. Materials

All chemicals were used as received unless otherwise stated. Ultrapure Milli-Q water, obtained with the help of a Labconco Water Pro PS (Kansas, KS, USA) purification system (18.2 MΩ), was used for the procedures. Poly(ethylene) tubing (0.56/1.07 mm inner/outer diameter) was purchased from Thermo Fischer (Waltham, MA, USA). An Eppendorf^®^ centrifuge 5430 R was used for filtration. An Eppendorf^®^ (Hamburg, Germany) ThermoMixer C was used for shaking. A Sylgard^®^ 184 silicone elastomer kit was used for the fabrication of the PDMS microfluidic chip. Trichloro(1H,1H,2H,2H-perfluorooctyl)silane (97%), dextran from *Leuconostoc mesenteroides* (average Mn 3500–45,000) and poly(ethylene glycol) diacrylate (PEGDA) (average Mn 575) were purchased from Sigma Aldrich (Saint Louis, MO, USA). Alginate methacrylate with a medium viscosity and a degree of methacrylation of 10–30% (ALMA) was purchased from AV Chemistry (AJ Nijmegen, The Netherlands). The photoinitiator 2-hydroxy-4′-(2-hydroxyethoxy)-2-methylpropiophenone (98%) (Irgacure 2959), 1H,1H,2H,2H-Perfluoro-1-octanol (97%), 5-FU (130.08 g/mol), and fluoresceinamine isomer I (347.32 g/mol) were bought from Sigma Aldrich. 008-FluoroSurfactant was purchased from RAN Biotechnologies (Beverly, MA, USA). 3M™ Novec™ 7500 Engineered Fluid was purchased from FluoroChem (Glossop, UK). N-Hydroxysulfosuccinimide sodium salt (sulfo-NHS) and 1-Ethyl-3-(3-dimethylaminopropyl)carbodiimide hydrochloride (EDC) were purchased from Fluorochem EU. Phosphate buffered saline was bought from Sigma-Aldrich. Dulbecco’s Modified Eagle’s Medium (DMEM), foetal bovine serum (FBS), penicillin/streptomycin, and trypsin/EDTA were purchased from Thermo Fisher. Cell Counting-8 (CCK-8 kit) was bought from Sigma-Aldrich. The µ-slide 8-well chambered coverslip was bought from Ibidi (Gräfelfing, Germany), calcein AM and propidium iodide (PI) where purchased from Sigma-Aldrich.

### 2.2. PDMS Microfluidic Device

A mixture of monomer and initiator (10:1 *w*/*w*) was poured onto the silicon master, after which it was degassed under vacuum for at least 4 h. The PDMS was cured at 65 °C overnight, washed with isopropanol, and blow-dried. After an oxygen plasma treatment, the PDMS was bonded to a glass slide. The channels were coated with trichloro(1H,1H,2H,2H-perfluorooctyl)silane (2% *w*/*w* in fluorinated oil) and the device was baked at 100 °C overnight.

### 2.3. PDMS Generation of Hydrogel Microparticles

PEGDA–ALMA microgels were prepared with both spherical and asymmetric shape by using a two- and three-inlet microfluidic device, respectively. For each morphology, the polymer mixture of the main aqueous phase was prepared by keeping constant the concentration of PEGDA (30% *w*/*w*), while the concentration of ALMA was varied between 0.75% *w*/*w* and 1.5% *w*/*w*. As a result of this, four samples of hydrogel microparticles with different morphologies were obtained.

#### 2.3.1. PEGDA–ALMA Spherical Microbeads

For the investigation of the influence of the shape on drug loading and release, microparticles with spherical morphology were generated in a two-inlet microfluidic device and their behaviour was compared to the one of anisotropic particles. The spherical microparticles were prepared from a main-phase aqueous solution in which the concentration of PEGDA was kept constant at 30% *w*/*w*, while the concentration of ALMA was either 0.75% *w*/*w* or 1.5% *w*/*w*. The solutions were prepared in a glass vial and kept for 4 h on a rolling bench. Afterwards, they were flushed with nitrogen for at least 0.5 h. The fluorocarbon oil (HFE 7500) was flushed for 15 min, to remove dissolved oxygen. Fusion 100 Touch pumps (KR Analytical Ltd.) were used for injecting the solutions in the channels of the microfluidic device. Irgacure^®^ 2959 (0.4% wt final concentration) was added to the main-phase solution prior to injection. The PEGDA–ALMA solution was injected in the second inlet, while the outer phase consisting of a mixture of fluorocarbon oil (HFE 7500) and surfactant (SS01, 2% *w*/*w*) was injected in the first inlet. Spherical droplets were formed and emulsified at the cross junction, and they were collected in an Eppendorf^®^ tube in which mineral oil (30 μL) was added to ensure the first particles did not break. Asahi spectra Max-300^®^ was used to photocure the hydrogel beads. The UV curing of PEGDA–ALMA was achieved by exposing the emulsion to a focused UV beam (λ = 320–500 nm, 360 s, 45% light intensity). The emulsion was broken by adding 1H,1H,2H,2H-Perfluoro-1-octanol (300 µL, 20% *w*/*w* in hexane), after which the beads were washed three times with Milli-Q water. During the washing, the sample was centrifuged at 20 °C and 14,000 rpm for 7 min, the supernatant was discarded and replaced with 500 µL of Milli-Q. During injection, the flowrates were 500 µL/h for the oil and 60 µL/h for PEGDA–ALMA.

#### 2.3.2. PEGDA–ALMA:Dextran Asymmetric Microparticles

PEGDA–ALMA:dextran asymmetric microgels were generated in a three-inlet microfluidic chip (Appendix A). In order to investigate the influence of different amounts of ALMA on the pH-responsive behaviour, the microparticles were prepared from a main-phase aqueous solution in which the concentration of PEGDA was kept constant at 30% *w*/*w*, while the concentration of ALMA was either 0.75% *w*/*w* or 1.5% *w*/*w*. The corresponding phase was a solution of 20% *w*/*w* dextran. All the solutions were prepared in a glass vial and kept for 4 h on a rolling bench. Afterwards, they were flushed with nitrogen for at least 0.5 h. The fluorocarbon oil (HFE 7500) was flushed for 15 min, to remove dissolved oxygen. Fusion 100 Touch pumps (KR Analytical Ltd.) were used for injecting the solutions in the channels of the microfluidic device. The PEGDA-ALMA and dextran solutions were injected in the second and third inlet, respectively. Irgacure^®^ 2959 (0.4% wt final concentration) was added to the main-phase solution prior to injection. The droplets were formed at the second cross junction by the introduction of an outer phase which consisted of a fluorocarbon oil (HFE 7500) and surfactant (SS01, 2% *w*/*w*). The resulting emulsion was collected in an Eppendorf^®^ in which mineral oil (30 μL) was added to ensure the first particles did not break. Asahi spectra Max-300^®^ was used to photocure the hydrogel beads. The UV curing of PEGDA–ALMA was achieved by exposing the emulsion to a focused UV beam (λ = 320–500 nm, 360 s, 45% light intensity). The emulsion was broken by adding 1H,1H,2H,2H-Perfluoro-1-octanol (300 µL, 20% *w*/*w* in hexane), after which the beads were washed three times with Milli-Q water. During the washing, the sample was centrifuged at 20 °C and 14,000 rpm for 7 min, and the supernatant was discarded and replaced with 500 µL of Milli-Q. The flowrates during injection were kept at 600 µL/h for the oil, 60 µL/h for PEGDA–ALMA, and 20 µL/h for dextran.

### 2.4. Experimental Buffers and Solutions

All the experimental buffers and solutions were prepared in Milli-Q. For the analysis of the microgels at lower pH, drug loading and release studies, a hydrochloric acid–potassium chloride buffer (0.1 M, pH 2.0) was used. For the investigation of the microgels’ mechanical properties and drug release studies, potassium phosphate (0.1 M, pH 6.7) and a tris-HCl buffer (0.1 M, pH 7.4) were used. After preparation, the pH was adjusted with NaOH (1M) and HCl (1M) and measured by a pH-meter (FiveEasy-Mettler Toledo B.V., JK Tiel, The Netherlands). The buffers at pH 2.0, 6.7, and 7.4 were used to mimic the pH of gastric fluid, intestinal fluid, and the colonic pH, respectively. The influence of calcium ions on the PEGDA–ALMA network was assessed by using a 1% *w*/*v*
${CaCl}_{2}$ solution.

### 2.5. Characterization of Microgels

#### 2.5.1. FT-IT Spectroscopy

Fourier transform infrared spectroscopy (FT-IR) spectral measurements were performed using an FT-IR (Shimadzu spirit-T, IR-ATR) spectrophotometer in the transmittance mode, connected to a PC and the data were analysed by LabSolutions IR 2.2 software. Spectra were scanned between 4000 and 500 cm^−^^1^. A solution of 30% *w*/*w* PEGDA in Milli-Q, a solution of 1.5% ALMA in Milli-Q, and a solution of the mixture PEGDA (30% *w*/*w*)–ALMA (1.5% *w*/*w*) were measured. To confirm the formation of the PEGDA–ALMA composite hydrogel, a solution of PEGDA (30% *w*/*w*)–ALMA (1.5% *w*/*w*) was flushed under nitrogen, then Irgacure^®^ 2959 (0.4% wt final concentration) was added and UV curing was achieved by exposing the solution to a focused UV beam (λ = 320–500 nm, 360 **s**, 45% light intensity). The resulting hydrogel was freeze-dried overnight, finely ground, and then measured.

#### 2.5.2. Cryo-SEM Analysis

Cryogenic scanning electron microscopy (Zeiss Crossbeam 550 (cryo) FIB-SEM) was used to investigate the change in morphological properties of the microgels after treatment with different solutions, namely buffer pH 2.0, buffer pH 7.4, and 1% *w*/*v*
${CaCl}_{2}$. After fabrication, every sample of microparticles was kept in a thermoshaker at 800 rpm and 20 °C for 30 min in each solution, then it was slush-frozen in liquid nitrogen, and finally, fractured. Prior to the cryo-SEM analysis, the cross sections of the freeze-dried hydrogel specimens were gold-coated.

#### 2.5.3. Rheology Studies

The mechanical properties were analysed on a stress-controlled rheometer (Discovery HR-2, TA Instrument) using a steel parallel plate geometry (EHP Steel-108195) with a plate diameter of 20 mm. The samples were loaded into the rheometer as thin-film hydrogels with a thickness of 2 mm. The solutions of PEGDA (30% *w*/*w*)–ALMA (0.75% or 1.5% *w*/*w*) were prepared in a glass vial and treated under nitrogen flow for 15 min. Irgacure^®^ 2959 (0.4% wt final concentration) was added and then the solutions were UV-photocured (λ = 320–500 nm, 360 s, 45% light intensity) in a steel mould with a diameter of 20 mm. Afterwards the thin-film hydrogels were immersed for 30 min in Milli-Q, pH 2.0 and pH 7.4 buffers, or a 1% *w*/*v*
${CaCl}_{2}$ solution, respectively. A time-sweep rheology analysis was performed to measure the storage modulus (G’) of the hydrogels at a constant temperature of 25 °C. All rheological measurements were conducted at constant strain and constant stress (636,620 Pa/N·m).

#### 2.5.4. Microgels Shrinking and Swelling Measurements

Shrinking and swelling studies of the microgels were performed in pH 2.0 and pH 7.4 buffers, in a 1% *w*/*v*
${CaCl}_{2}$ solution and in pH 2.0 and 7.4 buffers after treatment with 1% *w*/*v*
${CaCl}_{2}$. The pH responsiveness of both spherical and asymmetric microgels with a higher concentration of ALMA was also investigated in a pH 6.7 buffer solution. After fabrication, each sample was split into six (microgels ALMA 0.75% *w*/*w*) or seven (microgels ALMA 0.75% *w*/*w*) Eppendorf tubes with an equal volume (100 µL). Of those, one was kept in Milli-Q and the other five were centrifuged at 20 °C and 14,000 rpm for 7 min, then the supernatant was discarded and replaced by 500 µL pH 2.0, pH 6.7, pH 7.4 buffers and a ${CaCl}_{2}$ solution. The samples in the freshly added solutions were kept in a thermoshaker for 30 min at 800 rpm and 20 °C. Three aliquots of the samples were kept in ${CaCl}_{2}$ for 30 min, of which one was imaged without further modification in the environment solution, and the other two were centrifuged again, and the supernatant was discarded and replaced by either pH 2.0 or pH 7.4 buffers (500 µL). The microgels kept in different environment conditions were imaged with a Leica DMi8 widefield microscope (Leica Thunder) in bright-field mode and analysed using Fiji–ImageJ. On a glass slide, SecureSeal^®^ imaging spacers were attached to create a well. In the well, 30 μL of the microgels was added. The well was closed off be a coverslip. For the spherical microgels, the diameter (2R) of 50 particles in each solution was measured. For the asymmetric particles, both the full diameter (2R) and the respective cavity diameter (2r) of 50 microgels were measured. The results are plotted as mean ± standard deviation. The coefficient of variation (CV) was calculated as (standard deviation/mean) × 100. The volume of spherical particles was calculated according to Equation (1):(1)$$V_{\left(2R\right)}=\frac{4}{3}\pi ({\frac{2R}{2})}^{3}$$

The volume of asymmetric microgels was calculated as Equation (2), where $V_{\left(2R\right)}$ is the volume calculated from the full diameter of the particle and $V_{\left(2r\right)}$ is the volume calculated from the diameter of its corresponding cavity (Equation (2)):(2)$$V_{Asymmetric\;microgel}=V_{\left(2R\right)}-V_{\left(2r\right)}$$

The percentage occupied by the cavity ($V_{\left(2r\right)})$ on the full microgel ($V_{\left(2R\right)}$) was expressed as Equation (3).
(3)$${\%\;}_{V_{\left(2r\right)}:V_{\left(2R\right)}}=\frac{V_{\left(2R\right)}}{V_{\left(2r\right)}}\;*\;100$$

### 2.6. Fluorescence of PEGDA–ALMA Asymmetric Microgels

#### 2.6.1. Fluorescence Labelling of ALMA

ALMA was fluorescently labelled according to a procedure previously reported [51]. In total, 320 mg of ALMA was dissolved in PBS 0.01 M (NaCl 0.138 M; KCl 0.0027 M) pH 7.4, at 25 °C to give approximately 90 mM carboxylic groups. EDC (70 mg) and Sulfo-NHS (98 mg) were added to 9 mM of each. The solution was kept at 20 °C while stirring for 2 h. The fluoresceinamine isomer (78 mg) was added to concentrations of 4.5 mM and the reaction mixture was stirred in the darkness for 18 h at 20 °C. In order to remove the unreacted fluoresceinamine, series of dialysis were performed in the darkness. The solution was first transferred into dialysis membranes (MWCO 12,000–14,000, Medicell LTD, UK) and dialyzed against ion-free water at 4 °C (one shift), then dialyzed against 1M NaCl for 24 h (three shifts), and finally, against ion-free water until the water no longer showed a yellow colour (four shifts). The pH of the ALMA solution was adjusted to 7.4 with a PBS buffer, then the sample was freeze-dried overnight, protected from light. The fluorescence-labelled ALMA was stored at 4 °C in the darkness until further use.

#### 2.6.2. Fluorescently Labelled PEGDA–ALMA Asymmetric Microgels

The fluorescently labelled PEGDA–ALMA:dextran microgels were generated in a three-inlet microfluidic device according to the procedure described in Section 2.3.2. The microparticles were prepared from a main-phase aqueous solution of 30% *w*/*w* PEGDA and 1.5% *w*/*w* fluorescently labelled ALMA (Section 2.6.1). The corresponding phase was a solution 20% *w*/*w* of dextran.

#### 2.6.3. Fluorescent Images of PEGDA–ALMA Asymmetric Microgels

The PEGDA-fluorescently labelled ALMA asymmetric microparticles were imaged with a Leica Thunder microscope. The microscope was equipped with an environmental-control incubator that was kept closed during imaging to prevent exposure of the sample to light. On a glass slide, SecureSeal^®^ (Grace Bio-Labs, Bend, USA) imaging spacers were attached to create a well. In the well, 30 μL of the microgels was added. The well was closed off be a coverslip. A 20×/0.80 Air (0.40 mm) objective was used to visualize the sample, a Lumencor 4-line LED (excitation) was set at a wavelength of 470 nm and a DFT5 quad-cube (emission) for high-speed fluorescence imaging was set at 506–532 nm. The images were acquired with LAS X 3D software for a 3D rendering of the acquired volumes and Fiji-ImageJ was used to analyse the fluorescence intensity.

### 2.7. Loading and Release of 5-FU

The loading and release efficiency of both asymmetric and spherical PEGDA–ALMA microgels at either 0.75% *w*/*w* or 1.5% *w*/*w* ALMA were determined by high-performance liquid chromatography (HPLC). The HPLC measurements were performed on an Agilent AG1120 Compact HPLC (Agilent Technologies, Amstelveen, The Netherlands) equipped with a Prodigy column, 150 × 4.6 mm, with a particle size of 5 μm (Phenomenex, Utrecht, The Netherlands). The isocratic mobile phase was composed of 0.1% trifluoroacetic acid in water with an increasing gradient of acetonitrile (5–70%, 1–40 min., flowrate 1.0 mL/min). The retention time was 2–3 min. The calibration curve between 5–100 µg/mL was established at pH 2.0, pH 6.7, and pH 7.4.

#### 2.7.1. Cumulative Loading of 5-FU

After fabrication and washing, the microgels were centrifuged at 20 °C and 14,000 rpm for 7 min, all the supernatant was removed, and 1 mg of the sample was transferred to a new Eppendorf, to which 600 µL of the 5-FU solution (40 µg/mL final concentration in the sample) buffered at pH 2.0 was added. The drug loading was measured at several time points (10, 30, 60, 90, 240, 420, and 1440 min). During the loading, samples were kept in a thermoshaker at 20 °C and 900 rpm. At each time point, the samples were centrifuged at 20 °C and 14,000 rpm for 7 min, then 300 µL of the supernatant was taken for HPLC measurements, and 300 µL of the 5-FU solution (240 µg/mL) was added. The concentration (x) of the supernatant at each time point was calculated according to the equation y=a+bx, where y is the response (mAU·s), and a and b are, respectively, the intercept and slope of the standard curve of 5-FU at pH 2.0. The drug loading was performed for 24 h, and it was calculated as a cumulative amount (µg) of 5-FU loaded in 100 µg of empty microgels. The reported values are expressed as mean of three replicates (*n* = 3) ± the cumulative standard deviation for each time point. The drug loading capacity of microgels was calculated as Equation (4).
(4)$$loading\;capacity=\frac{cumulative\;amount\;of\;5FU\;loaded\;(µg)}{amount\;of\;microparticles\;(µg)}\;*\;100$$

#### 2.7.2. pH-Responsive Drug Release

After 24 h of drug loading, each sample was centrifuged at 20 °C and 14,000 rpm for 7 min, all the supernatant was discarded, and 500 µL of either pH 2.0, pH 6.7, or pH 7.4 buffer solution was added. While conducting the in vitro release experiments, the samples were kept in a thermoshaker at 37 °C at 900 rpm to ensure that the microparticles would not sink to the bottom of the Eppendorf tube and to favour the desorption of 5-FU. At each time point (10, 30, 60, 90 and 240 min), the samples were centrifuged at 37 °C and 14,000 rpm for 7 min, then all the supernatant was removed, from which 300 µL was used for the HPLC measurements. After removal of the supernatant, 500 µL of pH 2.0, 6.7, or 7.4 buffer solution was added to the sample to proceed with the drug release investigation at the next time point. The concentration (x) of the supernatant at each time point was calculated according to the equation y=a+bx, where y is the response (mAU·s), and a and b are, respectively, the intercept and slope of the standard curve of 5-FU at each respective pH. The cumulative percentage of drug release was calculated as the percentage of the cumulative amount of drug released at each time point per cumulative amount of 5-FU loaded in 100 µg of each sample after 24 h. The reported values are expressed as the mean of three replicates (*n* = 3) ± the cumulative standard deviation for each time point.

### 2.8. Cell Viability of Empty Microgels

#### 2.8.1. Cell Culture and Cytotoxicity Assay

Fibroblast cells (NIH/3T3, isolated from a mouse NIH/Swiss embryo) were cultured in Dulbecco’s Modified Eagle’s Medium containing 10% foetal bovine serum and penicillin/streptomycin (100 U/mL, 100 U/mL) at 37 °C in a humidified atmosphere containing 5% of ${CO}_{2}$. Once the cells reached 80–90% confluency, they were passaged by trypsinization using trypsin/EDTA. The fibroblast cells were treated with different concentrations of the four different kinds of microgels.

#### 2.8.2. Cytotoxicity Measurements

The microgels cytotoxicity effect was investigated using a Cell Counting Kit-8 (CCK-8) assay. NIH/3T3 cells were cultured into 96-well plates with a density of 1 × 10^4^ per well. After overnight incubation, once the cells reached 90% confluency, series of concentrations of particles (1, 2, 5, 10, 20, 50, 100, 200, and 500 µg/mL) were added into the wells. After 72 h, a CCK-8 work solution (10 µL) was added to each well, followed by a 4 h incubation. Finally, the absorbance was detected by a UV spectrophotometer at the wavelength of 450 nm. The data are presented as mean ± standard deviation (*n* = 5). Statistical significance was determined by a one-tailed t-test and it was established at the level of *p* < 0.001.

### 2.9. Cell Viability of 5-FU-Loaded Microgels

#### 2.9.1. Cell Culture and Cytotoxicity Assay

HeLa cells were cultured in Dulbecco’s Modified Eagle’s Medium supplemented with 10% foetal bovine serum and penicillin/streptomycin (100 U/mL, 100 U/mL) at 37 °C in a humidified atmosphere containing 5% of ${CO}_{2}$. Once the cells reached 80–90% confluency, they were passaged by trypsinization using trypsin/EDTA. Finally, they were treated with the four kinds of microgels loaded with different concentration of 5-FU.

#### 2.9.2. Cytotoxicity Measurements

The cytotoxicity effect of different concentrations of 5-FU loaded in the microgels was investigated using a Cell Counting Kit-8 (CCK-8) assay. HeLa cells were cultured into 96-well plates with a density of 1 × 10^4^ per well. After overnight incubation, once the cells reached 90% confluency, series of particles loaded with different concentrations of 5-FU (1, 2, 5, 10, 20, 50, 100 µM) were added into the wells. After 72 h, a CCK-8 work solution (10 µL) was added to each well, followed by a 4 h incubation. Finally, the absorbance was detected by a UV spectrophotometer at the wavelength of 450 nm. The data are presented as mean ± standard deviation (*n* = 3). Statistical significance was determined by a one-tailed t-test and it was established at the level of *p* < 0.01.

### 2.10. Live–Dead Assay

HeLa cells were seeded in a µ-slide 8-well chambered coverslip, at a density of 5×10^4^ cells per well. After overnight incubation, PEGDA 30% *w*/*w*–ALMA 1.5% *w*/*w* asymmetric microparticles were added to different wells. The final quantity of particles was 5, 27, and 108 mg to reach a final concentration of 5-FU of 10, 50, and 200 μM, respectively. HeLa cells were incubated with loaded microparticles at 37 ℃ for 24 h. After rinsing with PBS three times, cells were cultured in the presence of calcein AM (2 μM) and PI (4.5 μM) for 30 min in the dark. An Echo Revolution fluorescent microscope was employed to observe the survival of HeLa cells upon different treatments.

### 2.11. Statistical Analysis

The statistical differences between the investigated groups were determined using Student’s *t*-test. Statistically significant and very significant differences between sets were found at *p* < 0.01 and *p* < 0.001, respectively.

## 3. Results and Discussion

### 3.1. PEGDA–ALMA Cross-Linked Polymeric Microgels

Photo-cross-linked microgels offer the possibility to obtain DDS very stable over time. Our pH-responsive hydrogel microparticles were made by creating an aqueous solution that contained PEGDA and ALMA as macromers and Irgacure as the photoinitiator. A short exposure to low-intensity light and an appropriate concentration of photoinitiator allowed for the generation of soft gels, with ideal characteristics for biomedical applications. Upon UV light exposure, Irgacure generated free radicals that initiated the free-radical polymerization (FRP) of the difunctional PEGDA and the methacrylate function of ALMA, yielding hydrogels in a water environment. Due to the presence of only one acrylate photocurable moiety, ALMA could not be used just by itself to create robust microgels. Therefore, PEGDA was used as a cross-linking agent to contribute to the formation of a durable network [52]. The formation of covalent bonds between the diacrylate moieties of PEGDA and the acrylate function of ALMA was confirmed by FT-IR spectroscopy (Appendix A). The FTIR spectrum of a mixture of PEGDA and ALMA before cross-linking showed as expected a single broad intense peak in the range of 1500–1700 cm^−1^, which consisted of the carbonyl and (meth)acrylic double bonds from both polymers. During the cross-linking step with UV, the (meth)acrylic groups polymerized, thus the C=C double bonds were converted into a C-C bond. The FTIR spectrum of a mixture of PEDGA and ALMA after UV irradiation clearly showed a single sharp carbonyl ester peak at 1700 cm^−1^ as the C=C polymerized and thus disappeared. In addition, in the FTIR spectrum of PEGDA and ALMA after UV irradiation the C-H stretching peak for alkenes at 3100 cm^−1^ disappeared and the C-H bending vibration for alkanes at 1450 cm^−1^ was stronger, which further proved that the C=C double bonds had polymerized.

### 3.2. Microfluidics for the Generation of PEGDA–ALMA Microparticles

The versatility of microfluidics for the generation of asymmetric microparticles has already been reported by our group [53,54]. Taking advantage of this knowledge, we used a microfluidic device with three (Figure 1) and two inlets (Appendix A) to prepare asymmetric and spherical microgels, respectively (Appendix A). Both morphologies were fabricated by using a main aqueous mixture of PEGDA 30% *w*/*w* and ALMA either 0.75% *w*/*w* or 1.5% *w*/*w*. The spherical microparticles were formed at the cross junction of the two-channel chip, where the main composite polymer solution met the flow of surfactant in oil. The emulsification ensured that the one-phase droplets were homogeneous in shapes and sizes. After UV-photocuring, it was possible to break the emulsion and preserve the stable spherical microgels in the aqueous environment.

For the generation of asymmetric particles, the presence of a corresponding aqueous solution of dextran 20% *w*/*w* was needed to create an aqueous-two-phase separating system (ATPS) [55]. The main and corresponding phases were two immiscible aqueous solutions that formed a two-phase jet at the first cross junction of the three-inlet microfluidic device (Appendix A). As result of this, a droplet-in-droplet architecture was obtained and collected at the outlet. The PEGDA–ALMA templating phase underwent a polymerization during UV exposure, while the dextran phase diffused into it, leaving behind a cavity; thus, the resulting microbeads presented an anisotropic shape. Several ATPSs can be found in the literature [56]; however, not many involve the presence of a stimuli-responsive aqueous phase. It is even more difficult to fine-tune the properties, such as the solution’s viscosity, of an ATPS for a high throughput in microfluidics. Our strategy involved the presence of PEGDA in the main phase, that not only acted as cross-linking agent but also provided the requirements for the aqueous phase separation with the corresponding phase.

### 3.3. Morphological Characterization of Microgels

#### 3.3.1. Differences between Spherical and Asymmetric Microparticles

The morphological characterization of hydrogel microparticles is crucial to monitor their shape, surface texture, and internal structure. Cryo-SEM images showed details of the microgels network and highlighted the differences between spherical and asymmetric PEGDA 30% *w*/*w*–ALMA 1.5% *w*/*w* architectures (Figure 2). By fracturing the sample, it was possible to see the cavity that gave the anisotropic morphology (Figure 2b). This confirmed that the aqueous phase separation between main and corresponding phase occurred successfully, creating droplet-in-droplet emulsions, which, upon UV exposure, turned into asymmetric particles. At the interface of the two phases, there were some protrusions of dextran that, due to its high molecular weight, could not completely diffuse into the tight gel [57]. The polymer network and mesh appeared uniform within the structure, as evidence of the well-occurred FRP between acrylate moieties of PEGDA and ALMA. Unfractured microgel spheres (Figure 2d) displayed regular shapes and a uniformly cross-linked surface. In comparison to PEGDA 30% *w*/*w*–ALMA 0.75% *w*/*w* (Appendix A), microgels with a higher concentration of ALMA presented smaller pores. This demonstrated that the overall quantity of macromers in solution affected the cross-linking, thus resulting in a denser gel network for a higher total polymer concentration.

#### 3.3.2. Influence of pH on the Microgels Structure

The pH-dependent behaviour of ALMA, because of the presence of carboxyl groups in its backbone, conferred attractive properties to the microgels. The pK_a_ values of (1→4)-linked residues of β-d-mannuronic (M) and α-l-guluronic (G) acids are 3.38 and 3.65, respectively [58]. At pH 2.0, the carboxylic moieties were protonated, meaning that the gels shrunk due to the hydrogen bonding between −COOH and −OH. This appeared as a very smooth and nonporous structure (Figure 3a). At pH 7.4, the carboxylic groups were in the form of ${-COO}^{-}$ and, due to electrostatic repulsion, they made the gel expand, which was visible on the cryo-SEM images (Figure 3b) as a change in the microgels’ density, resulting in larger pores at higher pH. This phenomenon was reversible as the properties of the surrounding media changed; in fact, right after the fabrication and washing with Milli-Q, when the microparticles were in a pH 7.0 environment, they showed large pores within the network (Appendix A) that altered in the presence of different pH. In this work, we used the acrylate function of ALMA to create photo-cross-linked stable microgels that did not need the presence of calcium ions to form an ionic-cross-linked gel network. The investigation of a dual cross-linked ALMA-based system has already been reported [59], showing that the chelation due to ${Ca}^{2+}$ after photo-cross-linking did not yield any advantage, being nevertheless a drawback for drug delivery. We investigated the effect of a 1% *w*/*v*
${CaCl}_{2}$ on our microgels, and it was clear that the diffusion of calcium ions into the gel network did not occur uniformly, generating an anisotropic gel network with a stiffer network only on the outer layer of the particles (Appendix A). This meant that the egg-box structure [60] was formed where there was a direct contact with the surrounding media, but the inner gel remained very porous. As already reported in the literature, the formation of a 3D gel network due to the gelation of alginate with divalent cations is not homogeneous and hard to control [61,62].

### 3.4. Mechanical Properties

Changes in mechanical properties of the microgels were confirmed by rheology studies. The storage modulus (G’) decreased as pH increased, showing that the hydrogel stiffness was lower due to swelling (Figure 3c). The electrostatic repulsion between carboxylate groups resulted in an enhancement of the expanding capacity, which meant a higher water content in the gel network. This was visible as larger pores in the cryo-SEM images and as less resistance to deformation of the material in the rheology measurements. For a lower concentration of ALMA in the network composition, there were less carboxylic moieties giving contribution to the pH-responsive behaviour of the hydrogel. This explained the small difference as a function of pH in the storage modulus of the PEGDA 30% *w*/*w*–ALMA 0.75% *w*/*w* microgels (Appendix A). The influence of ${Ca}^{2+}$ for the formation of egg-box structures after photo-crosslinking was also evaluated in terms of stiffness. For the microgels with a higher concentration of ALMA, a minor increase in the storage modulus could be observed after treatment with calcium chloride. The increase in G’ became slightly more evident when the ALMA concentration was lower. This could be explained considering that, due to the lower cross-linking density for a lower total starting concentration of polymers solution, the final network possessed bigger pores. The diffusion of divalent cations in a more porous material was faster and deeper into the core of the 3D structure, resulting into more “zipping” between ALMA G-blocks.

### 3.5. Swelling Measurements

PEGDA–ALMA microgels represent a potential DDS for oral administration of 5-FU in colorectal cancer. As such, it is crucial to investigate the pH responsiveness of microparticles in pH environments that mimic the gastrointestinal transit. Considering this, microgels were exposed to a pH 2.0 solution which simulated the gastric fluid, pH 6.7 for the intestinal fluid, and pH 7.4 corresponding to a colonic environment. The influence of the morphology and ALMA concentration on the swelling properties at different pH was investigated by keeping the microparticles in aqueous media. This ensured that each microgel was in its fully solvate state, so changes in size could be accounted only to the different charges in the environment. The measurements of the full diameter (2R) of both spherical and asymmetric morphologies showed that the anisotropic shape was more responsive to variations of the pH surroundings (Figure 4a). The 2R value of spherical particles with a higher concentration of ALMA was 27.6 ± 1.9 µm in Milli-Q (pH 7.0), and it decreased to 0.8 µm at pH 2.0. For asymmetric shapes, the value of 2R was 26.9 ± 2.5 µm in Milli-Q, it became 5.3 µm smaller at pH 2.0, and with subsequent treatment at pH 6.7 and 7.4, it increased again up to 23.7 ± 1.7 µm and 25.7 ± 2.5 µm, respectively. The anisotropy due to the cavity made the microgel more dynamic and responsive to the electrostatic repulsion. The spherical particles were more rigid because of the homogenous amount of polymer within the structure, whose core was less exposed to aqueous media. The anisotropy of asymmetrical shapes was confirmed by a small variation in size of the cavity (2r), compared to the evident change of 2R. At pH 2.0, 2R decreased to about 5.3 µm, while 2r only decreased to 1.9 µm (Figure 5a). By calculating the percentage of the volume occupied by the cavity ($V_{\left(2r\right)}$) in the total volume of the microgel ($V_{\left(2R\right)}$), we determined where in the structure of microgels the main shrinking happened. At pH 7.0, the cavity occupied 44% of the total volume, which became 60% at lower pH (Figure 5b). Therefore, changes in volumes occurred mainly in the shell of asymmetric particles, where there was a higher concentration of pH-responsive polymer. At the interface, due to the diffusion of dextran in the templating phase, there were less carboxylic moieties from ALMA that could be protonated. Due to its high MW, the dextran partly diffused into the main phase and partly remained in the cavity, visible as a rough texture in the bright-field images of asymmetric microgels (Figure 6, top). The 2R and 2r measurements of asymmetric particles after treatment with ${CaCl}_{2}$ showed once more that the interaction with divalent cations was not relevant for the shrinking behaviour (Appendix A).

### 3.6. Fluorescence Images

The anisotropic behaviour of PEGDA–ALMA:dextran microparticles depended on both the asymmetric shape and the ALMA distribution within the gel network. To visualise the location of pH-sensitive moieties, ALMA was fluorescently labelled. The fluorescence in the microgels, indicated as grey values, showed the homogenous organization of ALMA among the PEGDA chains (Figure 7), suggesting that there was no partitioning between the two polymers of the main aqueous templating phase before and after polymerization. Grey values for the side view of a microgel were higher in the shell, while they decreased at the interface (Figure 7b). This confirmed that the diffusion of dextran in the main phase upon UV exposure prevented a high cross-linking density between PEGDA and ALMA. As a result, the cavity was less responsive to pH changes due to the smaller amount of ALMA.

### 3.7. Cumulative Drug Loading

The loading of drugs into particles can be done either during fabrication, or afterwards. When it happens during fabrication, it is possible to achieve 100% of drug loading efficacy; however, there are some drawbacks that influence the performance of DDS. First, there might be an eventual damage of the drug molecules due to the reaction conditions, and second, the formation of covalent bonds between drug molecules and reactive groups on the polymer’s chains might occur. Moreover, the stability of many therapeutics is affected over time, compromising safety and efficacy. Developing DDS which adsorb drugs after fabrication implies the main advantage of on-demand loading. Our microgels were very robust and did not undergo chemical or structural deterioration over time, making them suitable for on-demand loading of drugs. The chemical stability of the hydrogel network was given by the formation of covalent C-C bonds between the diacrylate moieties of PEGDA and the methacrylate function of ALMA upon FRP. These bonds are very difficult to break unless the atoms are activated [63], thus even exposure to extreme pH conditions would not destabilize the microgels structure. For this reason, photo-polymerized PEGDA–ALMA microparticles represent better pH-responsive DDS compared to particles in which the gel network formation occurs because of ionic bonding between atoms of different polymers chains. In fact, the cleavage of ionic bonds would occur upon exposure to either very low or very high pH values depending on the pKa of functional groups, leading to damage of the particles structure. To evaluate the capabilities of our PEGDA–ALMA microgels as on-demand DDS, 5-FU loading measurements were performed over time in aqueous media at pH 2.0, mainly for two reasons. The first one is that 5-FU is a neutral weak acid [64] with a pKa = 7.76–8.02 [65], thus the formation of an anion can prevent its intercalation into the PEGDA–ALMA matrix, and the second is the existence of negatively charged carboxylic groups of ALMA above pH 3.0. The acidic environment avoids the electrostatic repulsion preventing the diffusion of 5-FU into the microgels and maximises the adsorption of 5-FU due to its great hydrogen bonding capacity, which was found to be optimal in aqueous solution at pH = 2.0–5.0 [66]. The asymmetric particles with 1.5% *w*/*w* ALMA adsorbed more than 50% of the total amount of 5-FU within 90 min, reaching a maximum of 24 µg in 24 h. This showed that the loading capacity was 0.24% *w*/*w*. The spherical counterpart could load a maximum of 12 µg in 24 h (Figure 8a), meaning a loading efficacy of 0.12% *w*/*w*. The drug adsorption was double for asymmetric particles: their anisotropic shape offered more surface area in contact with 5-FU, thus increasing the possibility for drug molecules to interact with the gel network. Another crucial factor influencing the loading was the higher pH responsiveness of the microgels. When the molecules of 5-FU were trapped into a tighter gel a low pH, it was more difficult for them to diffuse out of the particles. This explained the drug adsorption trend in particles with a lower concentration of ALMA and in spherical particles with 1.5% *w*/*w* ALMA. Finally, less hydrogen bonds could be formed between 5-FU and the hydrogels when the concentration of ALMA was lower, resulting in a reduced drug load (Appendix A).

### 3.8. In Vitro pH-Triggered 5-FU Release

Considering the employment of 5-FU as a first-line treatment for colorectal cancer, in vitro release studies of 5-FU were conducted at 37 °C in pH 2.0, 6.7, and 7.4 environments to recreate the pH values of the gastrointestinal tract and evaluate the performance of PEGDA–ALMA microgels as oral delivery system [67]. pH-responsive carriers, other than offering the possibility to achieve a targeted release at locations with a certain pH, give the advantage of protecting the stability of drugs from enzyme degradation or hydrolysis reactions that limit the effective delivery [68]. This implies that is not necessary to use a prodrug to improve the drug delivery to specific cells or tissues. In vitro studies showed a burst release effect of 5-FU from both spherical and asymmetric PEGDA–ALMA microgels in pH 7.4 media within the first 30 min (Figure 8b). This effect was more pronounced for the anisotropic particles, which also showed a higher rate of drug release over time. Within 90 min, almost 90% of 5-FU was released from asymmetric hydrogels, reaching 100% in 4 h, while at the same time point, the spherical particles had released only 60% of the adsorbed drug. At pH 7.4, the carboxylic groups of ALMA became negatively charged and the hydrogen bonding interactions with 5-FU decreased, starting the desorption of the drug molecules. The enlargement of the gel network pores due to the alkaline pH made the diffusion of 5-FU into the surrounding easier, while the polymer chains’ relaxation ensured more interactions with the medium. The less pH-responsive behaviour of the spherical particles with a higher concentration of ALMA due to the isotropy of the polymeric matrix played a big role in the percentage of drug released. Considering the lower loading efficacy and the release trend, spherical microgels released only 7.2 µg of drug, against the 24 µg released by the asymmetric counterpart. The almost six times higher performance of the asymmetric microgels as DDS confirmed that their anisotropic properties were crucial for the loading and pH-triggered release of drugs. The burst release profile of 5-FU forming both spherical and asymmetric particles with a lower ALMA concentration explained the fast diffusion of the drug out of the microgel network (Appendix A). At pH 7.4, the hydrogen bonds that kept a low amount of 5-FU adsorbed on the surface of the particles could not do that anymore due to the deprotonation of ALMA carboxylic moieties, favouring the desorption of the drug. These results showed that PEGDA–ALMA anisotropic microgels were an ideal oral drug delivery system for a colon-specific local action. In addition, it was observed that in pH mimicking the small intestine, the quantity of 5-FU released was only 2% after one hour and it increased up to a maximum of 21% and 27% after 4 h for PEGDA 30% *w*/*w*–ALMA 1.5% *w*/*w* asymmetric and spherical particles, respectively. This was in line with the less pronounced swelling behaviour of the microgels at pH 6.7, which led to a minor desorption of the drug. In a simulated gastrointestinal environment, in vitro experiments did not show any release. Overall, considering the influence of the morphology in combination with the weight percentage of pH-sensitive polymer, the asymmetric PEGDA 30% *w*/*w*–ALMA 1.5% *w*/*w* microgels had the best performance as oral DDS during both loading and release processes.

### 3.9. Cytotoxicity and Live/Dead Assay

In order to evaluate the cytotoxicity of the designed microgels, the viability of NIH/3T3 cells was measured using a CCK-8 assay after 72 h incubation with different concentrations of empty microparticles. Considering the biocompatibility of both PEGDA and ALMA polymers, it was not surprising that the cells’ viabilities remained higher than 80% in the range of concentrations investigated for three out of the four morphologies (Appendix A). Only the spherical PEGDA 30% *w*/*w*–ALMA 1.5% *w*/*w* particles for a concentration of 500 µg/mL gave a 77.5% cell viability, which was statistically significant compared to the 88.5% cell viability given by the asymmetric counterpart at the same concentration (Figure 9a). The 3D microstructure of hydrogel microparticles presented different mechanical properties due to the polymer composition and the anisotropic behaviour. The last one, in particular, affected the cells’ behaviour. It was postulated that spherical particles provided a less healthy and less favourable three-dimensional environment for the vital functions of the cells due to their stiffer network from a more cross-linked gel combined with a smooth surface and the isotropy given by the morphology.

The cytotoxicity of 5-FU-loaded microgels was studied on HeLa cells (Figure 9b). The amount of each kind of particles used for investigating the cell viability was calculated to obtain a final concentration of loaded 5-FU of 1, 2, 5, 10, 20, 50, and 100 µM. The results showed that after treatment with asymmetric microparticles, only 34% of the cells were alive, against 53% of cells alive after incubation with 5-FU-loaded spherical particles. This was explained by the release profiles, in which we observed 100% and 60% of released 5-FU by the asymmetric and spherical microgels, respectively. Finally, the live/dead assay showed that a 5-FU concentration of 200 µM loaded in PEGDA–ALMA asymmetric particles did not allow cancer cells to divide (Figure 10c).

## 4. Conclusions

The design of a microgel with an anisotropic behaviour can streamline the use of microparticles as DDS. In this study, we reported a novel, pH-responsive, asymmetric hydrogel system for an increased efficacy of 5-FU loading and release. Microfluidics was used for a high throughput of the microgels, whose templating phase was a mixture of two biopolymers, namely, PEGDA and ALMA. ALMA was used as the pH-responsive component, while its methacrylate function allowed for a photochemical cross-linking with the diacrylate PEGDA, leading to the formation of a stable network over time useful for drug delivery purposes. The influence of pH on the mechanical properties was investigated on spherical and asymmetrical particles made with two different concentrations of ALMA. It was found that the particles with 0.75% *w*/*w* ALMA did not show any pH responsiveness, while those with 1.5% *w*/*w* ALMA were very sensitive to changes in the pH environment. The most crucial factor on their performance as DDS was the morphology: microgels with anisotropic shape and properties displayed an enhanced drug adsorption and release. Cell viability studies suggested that empty asymmetric particles provided a more favourable 3D environment for cells’ growth, differentiation, and proliferation, while cell viability studies with 5-FU-loaded particles confirmed that, due to a higher release of drug, asymmetric microgels were more efficient as DDS for killing tumour cells. Our PEGDA–ALMA:dextran asymmetric hydrogel microparticles are promising on-demand drug-loading systems with an increased release efficacy. Further investigation could lead to a more in-depth understanding of the role of the surface area of asymmetric microgels as a matrix for the growth of cells with pH-tuneable mechanical properties.

## Figures and Tables

**Figure 1 pharmaceutics-15-01380-f001:**
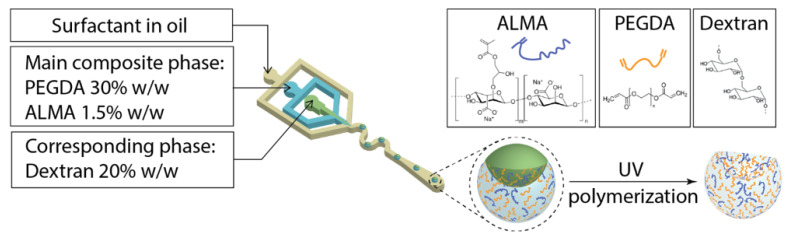
Schematic representation of the three-inlet microfluidic device used for the generation of asymmetric PEGDA–ALMA:dextran microgels. Droplet-in-droplet microparticles become asymmetric microgels after UV-photopolymerization.

**Figure 2 pharmaceutics-15-01380-f002:**
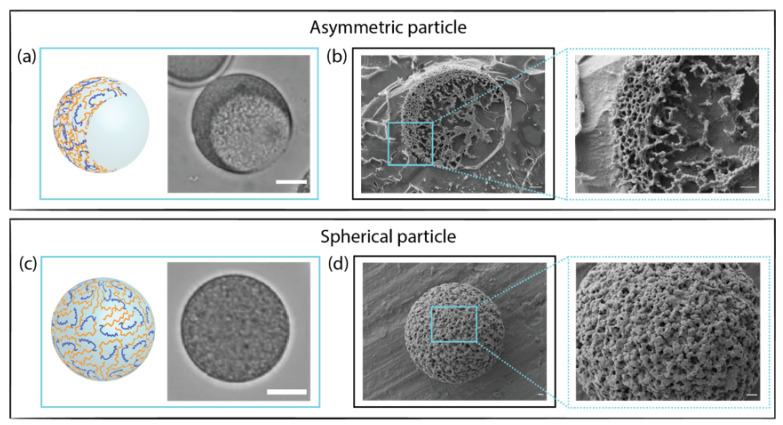
PEGDA 30% *w*/*w*–ALMA 1.5% *w*/*w* asymmetric and spherical microparticles. (**a**,**c**) Bright field images: scale bar is 10 µm. (**b**,**d**) Cryo-SEM images: scale bar is 1 µm.

**Figure 3 pharmaceutics-15-01380-f003:**
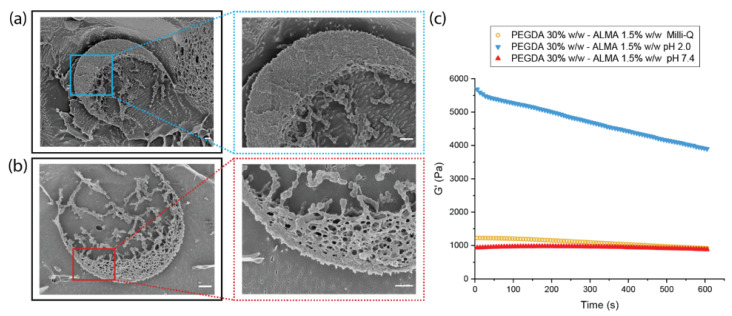
Cryo-SEM images of PEGDA 30% *w*/*w*–ALMA 1.5% *w*/*w*: dextran 20% *w*/*w* asymmetric microparticles in (**a**) pH 2.0 and (**b**) pH 7.4. Scale bar is 1 µm. (**c**) Storage modulus (G’) over time of PEGDA 30% *w*/*w*–ALMA 1.5% *w*/*w* hydrogel after treatment with different pH solutions.

**Figure 4 pharmaceutics-15-01380-f004:**
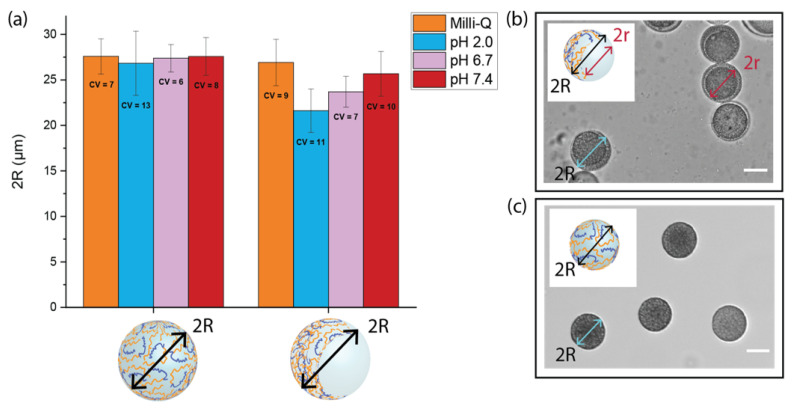
(**a**) Shrinking and swelling measurements of spherical and asymmetric PEGDA–ALMA microgels. The 2R symbol represents the microparticles diameter at different pH and is expressed as mean ± standard deviation (*n* = 50). CV is the coefficient of variation. Bright-field images of PEGDA–ALMA: (**b**) asymmetric and (**c**) spherical microgels in Milli-Q. Scale bar is 20 µm.

**Figure 5 pharmaceutics-15-01380-f005:**
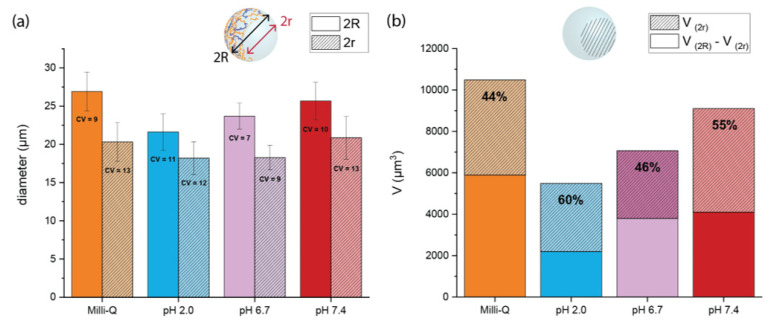
(**a**) Shrinking and swelling measurements of asymmetric PEGDA–ALMA:dextran microgels at different pH. The 2R symbol represents the full diameter of the particle and 2r is the diameter of its respective cavity. Values are expressed as mean ± standard deviation (*n* = 50). CV is the coefficient of variation. (**b**) Percentage of the volume occupied by the cavity in the full volume of the PEGDA–ALMA:dextran microgels. Values are averaged over 50 particles.

**Figure 6 pharmaceutics-15-01380-f006:**
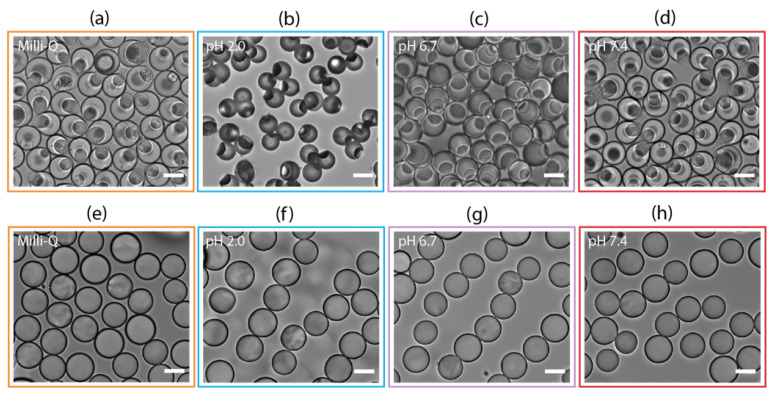
Bright-field images of asymmetric (**a–d**) and spherical (**e–h**) PEGDA 30% *w*/*w*–ALMA 1.5% *w*/*w* microgels at different pH. Scale bar is 20 µm.

**Figure 7 pharmaceutics-15-01380-f007:**
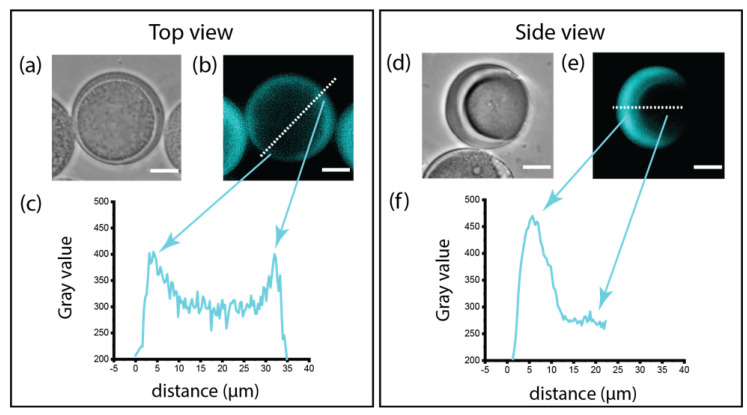
Top and side views of an PEGDA–ALMA:dextran asymmetric microparticle (**a**,**d**) bright-field and (**b**,**e**) fluorescence images. Scale bar is 10 µm. The fluorescence intensity expressed as grey values, shows the homogeneous distribution of ALMA in the microgel’s shell (**c**), while less ALMA is located at the interface (**f**), due to the presence of dextran.

**Figure 8 pharmaceutics-15-01380-f008:**
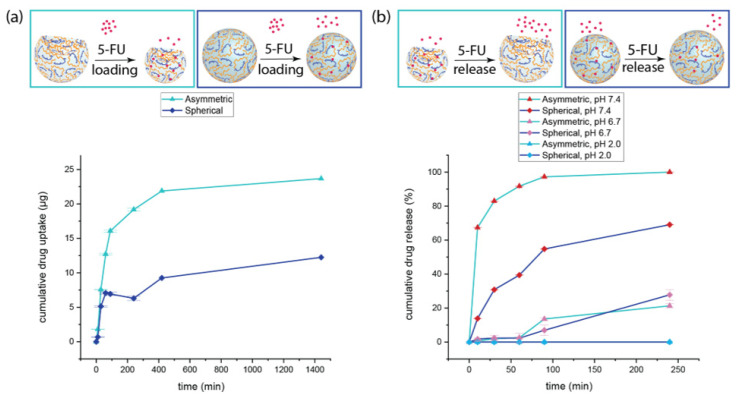
5-FU cumulative loading (**a**) and cumulative percentage of release (**b**) of asymmetric and spherical PEGDA–ALMA hydrogel microparticles. Values are expressed as mean of three replicates (*n* = 3) ± the cumulative standard deviation at each time point.

**Figure 9 pharmaceutics-15-01380-f009:**
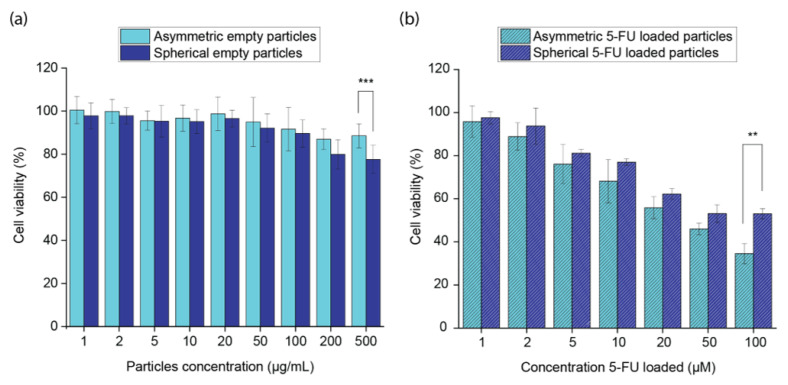
(**a**) NIH/3T3 cells’ viability after 72 h of incubation with different concentrations of PEGDA 30% *w*/*w*–ALMA 1.5% *w*/*w* empty spherical and asymmetric particles. The cell viability values are statistically significant different at 500 µg/mL. Values are plotted as mean ± standard deviation (*n* = 5), for each concentration. Statistical significance: *** is established for *p* < 0.001. (**b**) HeLa cells’ viability after 72 h of incubation with different PEGDA 30% *w*/*w*–ALMA 1.5% *w*/*w* spherical and asymmetric particles loaded with different concentrations of 5-FU. The cell viability values are statistically significant different at 100 µM. Values are plotted as mean ± standard deviation (*n* = 3), for each concentration. Statistical significance: ** is established for *p* < 0.01.

**Figure 10 pharmaceutics-15-01380-f010:**
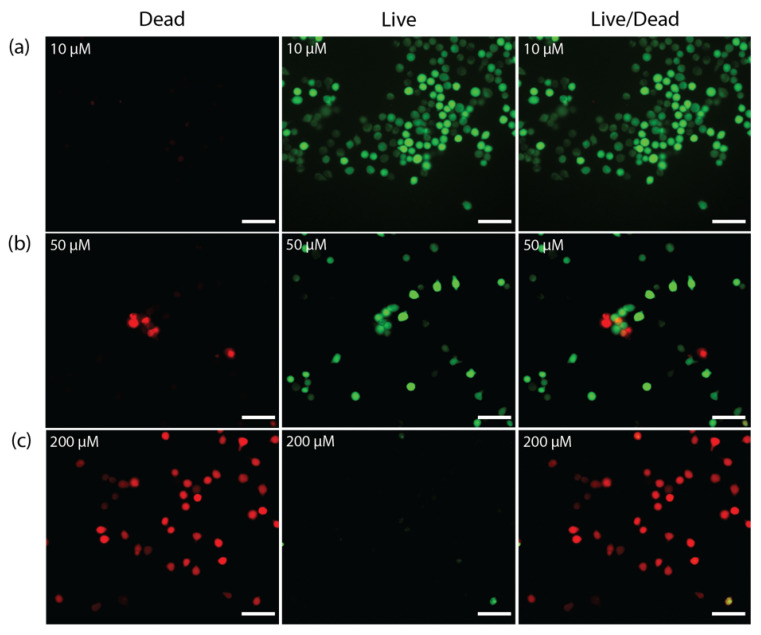
Live/dead assay staining of HeLa cells with PEGDA 30% *w*/*w*–ALMA 1.5% *w*/*w* asymmetric microgels loaded with a final 5-FU concentration of (**a**) 10 µM, (**b**) 50 µM, and (**c**) 200 µM. Green cells are living cells and red ones are dead. Scale bar is 50 µm.

## Data Availability

Data are contained within the article or supplementary material.

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
