# Peer review of "Anisotropic, Hydrogel Microparticles as pH-Responsive Drug Carriers for Oral Administration of 5-FU"

_pharmaceutics, 2023, doi:10.3390/pharmaceutics15051380_

Round 1
Reviewer 1 Report
The manuscript by Serena et al. describes the development of pH-responsive PEGDA-ALMA based microgels for 5-fluorouracil (5-FU) loading. Overall, the manuscript is well written and organized. However, after reading the paper, I have the impression that it did a lot of material characterization while lack in vitro and in vivo studies to evaluate the biological activity of the microgels. As such, the manuscript is more suitable for a material journal rather than a pharmaceutics journal. The authors are encouraged to add more biological studies to support the efficacy of their formulation.
To improve the manuscript, the authors should carefully review the text to ensure there are no errors such as the ones found on lines 535 and 465. Additionally, there are inconsistencies with the figure numbering (two figure 1 and no figure 8), and the figures should be cited in order, like figure S1-S3.
For clarity, the section 3.7 on cumulative drug uptake is recommend renamed "loading" instead of "uptake".
The manuscript should clarify the intended administration route for their formulation, as the size of the microgels may not be suitable for IV administration. Given that 5-FU is typically administered as a slow IV push, if the authors intend to administer their formulation orally, they should consider using a prodrug like capecitabine instead.
Section 3.8 is titled "In Vitro pH-Triggered 5-FU Release," but the authors only studied the drug release in pH 7.4. To make the study more comprehensive, the authors should investigate the drug release profile at different pH levels.
The authors only studied the cytotoxicity of their microgels in NIH/3T3 fibroblast cells, and the study should be expanded to include microgel loading with 5-Fu in cancer cells. Doing so would strengthen the manuscript's biological significance and make it more suitable for a pharmaceutics journal.

Author Response
Response to the reviewer:
The manuscript by Serena et al. describes the development of pH-responsive PEGDA-ALMA-based microgels for 5-fluorouracil (5-FU) loading. Overall, the manuscript is well-written and organized. However, after reading the paper, I have the impression that it did a lot of material characterization while lacking in vitro and in vivo studies to evaluate the biological activity of the microgels. As such, the manuscript is more suitable for a material journal rather than a pharmaceutics journal. The authors are encouraged to add more biological studies to support the efficacy of their formulation.
To improve the manuscript, the authors should carefully review the text to ensure there are no errors such as the ones found on lines 535 and 465. Additionally, there are inconsistencies with the figure numbering (two figure 1 and no figure 8), and the figures should be cited in order, like figure S1-S3.
For clarity, the section 3.7 on cumulative drug uptake is recommend renamed "loading" instead of "uptake".
We thank the referee for pointing this out, we agree it is more appropriate to rename “loading”. As suggested the section has been renamed and the word “uptake” has been changed with “loading” in all the manuscript and figures to avoid confusion. All the figures have been also re-numbered progressively and cited in order in the manuscript.
The manuscript should clarify the intended administration route for their formulation, as the size of the microgels may not be suitable for IV administration. Given that 5-FU is typically administered as a slow IV push, if the authors intend to administer their formulation orally, they should consider using a prodrug like capecitabine instead.
The authors would like to thank this referee for this valuable comment. Considering the particles size, it was implied that they would be suitable for oral administration, but we agree that for clarity it is better to further specify. We explained in section 3.5 (line 489 – 493) the intention to use our system as oral DDS. In section 3.8 (line 574 – 581, 602 -609) we explained the reason why instead of using a pro-drug, our pH-responsive DDS allows for loading and efficient delivery of 5-FU.
Section 3.8 is titled "In Vitro pH-Triggered 5-FU Release," but the authors only studied the drug release in pH 7.4. To make the study more comprehensive, the authors should investigate the drug release profile at different pH levels.
The authors acknowledge the referee for this valuable suggestion. Extra in vitro release studies at different pH, namely 2.0 and 6.7 have been performed with both spherical and asymmetric PEGDA 30% w/w – ALMA 1.5 w/w microgels. The results are shown in figure 8b and discussed in section 3.8.
The authors only studied the cytotoxicity of their microgels in NIH/3T3 fibroblast cells, and the study should be expanded to include microgel loading with 5-Fu in cancer cells. Doing so would strengthen the manuscript's biological significance and make it more suitable for a pharmaceutics journal.
We thank the reviewer for this constructive suggestion. We have performed cytotoxicity experiments on HeLa cells with both asymmetric and spherical particles loaded with 5-FU. The results are presented in figure 9b, figure S14 supporting information and discussed in the manuscript in section 3.9. In addition, a live/dead assay has been performed and the images of the results are shown in figure 10.
Reviewer 2 Report
This manuscript developed a asymmetric PEGDA-ALMA hydrogel with the drug-carrying capacity of 5-FU and the ability to release the drug at pH 7.4. The results are interesting, and might be helpful for the development of intelligent hydrogel particles. But some experimental designs and title in this study is questioned. In general, pH-Responsive refers to the release of a drug in response to conditions different from physiological pH (7.4). In particular, the drug 5-FU mentioned in the article is an antitumor drug, so the pH chosen for drug release should be chosen to use a pH close to the tumor microenvironment (~6.5) for drug release experiments. Therefore, I cannot accept current manuscript for publication.
Author Response
This manuscript developed an asymmetric PEGDA-ALMA hydrogel with the drug-carrying capacity of 5-FU and the ability to release the drug at pH 7.4. The results are interesting, and might be helpful for the development of intelligent hydrogel particles. But some experimental designs and title in this study is questioned. In general, pH-Responsive refers to the release of a drug in response to conditions different from physiological pH (7.4). In particular, the drug 5-FU mentioned in the article is an antitumor drug, so the pH chosen for drug release should be chosen to use a pH close to the tumor microenvironment (~6.5) for drug release experiments. Therefore, I cannot accept current manuscript for publication.
We acknowledge the referee for this comment, however, we respectfully disagree with it. In this work, we present hydrogel microparticles that, due to their size, cannot be used for IV administration. Considering that they are more suitable for oral administration and 5-FU is widely used for colon cancer, we investigated the drug release at pH 2.0, 6.7, and 7.4 to simulate the GIT transit, where pH 7.4 corresponds to the pH of colonic environment.
Reviewer 3 Report
In this manuscript, Teora et al. describe the use of methacrylate-functionalized alginate in combination with PEGDA to prepare microgels by photochemical polymerization. Using the droplet microfluidic technique, two different types of microspheres are synthesized, namely spherical and asymmetric morphologies. Moreover, the influence of the shape and different concentrations of the pH-reactive polymer within the gel network on their performance as drug delivery systems (DDS) and their cytotoxicity is investigated. This manuscript is very well written, and I really enjoyed reading it.
Major concerns:
1. Two pH conditions are considered for shrinkage and swelling studies (pH 2 and 7.4). Why are these two values chosen? What is the effect of a higher base value on swelling? Does it destabilize the hydrogel beads? An explanation is required.
2. In the synthesis of single and double emulsions, flow rates are a trade-off parameter to manipulate the size, especially in double emulsions one can change the diameter of the shell and the core. Could the authors investigate different flow rates and the effects of a thicker and thinner shell on the release of 5-FU? Data on flow rates as a function of the diameter of the single emulsion and the thickness of the shell and core would improve readability.
3. In figure 3 (b and c), the interior of the spheres does not seem to be clear. Are there monomers on the chip that are encapsulated with the aqueous phase formed before polymerization? I suggest adding microscopic images of the droplet formation or even videos to know how the droplets are formed.
4. The coefficient of variation (CV) is an important parameter for the uniformity of the particles. State the CV value. Specify the particle density, and I would be happy to see the dense packing of single and double emulsion microspheres.
5. Why are no microscope images shown in the study of swelling and shrinkage? This is one of the most important aspects of this study. An image of swelling and shrinkage with the control should be included in the manuscript.
6. In the previous reports of the same group (Teora et al., Chem. Commun. 2022, 58, 10333, a thermosensitive hydrogel was synthesized based on poly(ethylene glycol) (PEGDA) 25 wt%, PNIPAm 10 wt%, and dextran 20 wt%. I am curious why 30% PEGDA is used in this study? On what basis was it selected and how does it affect yield and mechanical and physical properties?
7. In section 3.7, the maximum drug uptake of 12 µg in 24 hours is determined and a double efficacy is found for anisotropic particles. Can the authors provide the relationship between drug-loaded particles (number of particles) and cell viability? Just wanted to know how the quantification was done. How many particles are needed in the sample to uptake 12 µg in 24 hours?
Minor comments:
1. In the manuscript, the word “CaCl2 is bold and italics on various pages.” Make it consistent.
2. Line 424, Calcium ions to calcium ions.
3. Figure 2 C, the resolution needs to be improved.
4. Line 543, ~ 60%, please make it correct, the sign “~” is in the superscript.
Author Response
In this manuscript, Teora et al. describe the use of methacrylate-functionalized alginate in combination with PEGDA to prepare microgels by photochemical polymerization. Using the droplet microfluidic technique, two different types of microspheres are synthesized, namely spherical and asymmetric morphologies. Moreover, the influence of the shape and different concentrations of the pH-reactive polymer within the gel network on their performance as drug delivery systems (DDS) and their cytotoxicity is investigated. This manuscript is very well written, and I really enjoyed reading it.
Major concerns:
- Two pH conditions are considered for shrinkage and swelling studies (pH 2 and 7.4). Why are these two values chosen? What is the effect of a higher base value on swelling? Does it destabilize the hydrogel beads? An explanation is required.
We thank the referee for these questions. We added in the manuscript an explanation for the choice of pH in section 3.5 (line 489 – 493) and in section 3.7 (line 542 – 551) the reason why a higher pH would not destabilize the hydrogel network.
- In the synthesis of single and double emulsions, flow rates are a trade-off parameter to manipulate the size, especially in double emulsions one can change the diameter of the shell and the core. Could the authors investigate different flow rates and the effects of a thicker and thinner shell on the release of 5-FU? Data on flow rates as a function of the diameter of the single emulsion and the thickness of the shell and core would improve readability.
The authors acknowledge the referee for the interesting question. We do agree that the different flow rates for the injection of solutions in the microfluidic chip would generate morphologies with different thicknesses of shell and sizes of the cavity. Nevertheless, we previously reported how the flowrate ratio between the main phase solution and the corresponding phase solution would influence the size of the opening (High-Throughput Design of Biocompatible Enzyme-Based Hydrogel Microparticles with Autonomous Movement, DOI: 10.1002/anie.201805661). In this work, we use a 3:1 ratio in order to ensure the maximum anisotropy of the resulting microgels compared to the isotropic spherical particles. By reducing the size of the opening, we expect that the mechanical properties of asymmetric microgels would resemble those of spherical particles. Considering the amount of data presented in this work due to the comparison of two morphologies in combination with two concentrations of ALMA, for a total of four microgels, of which the behavior as DDS needs to be investigated at different conditions, we think that adding more results on the influence of shell thickness would rather increase the complexity of the manuscript.
- In figure 3 (b and c), the interior of the spheres does not seem to be clear. Are there monomers on the chip that are encapsulated with the aqueous phase formed before polymerization? I suggest adding microscopic images of the droplet formation or even videos to know how the droplets are formed.
We thank the referee for this suggestion. We added a video recorded during the generation of asymmetric microparticles in the supporting information (video 1).
In figure 4b most of the particles imaged show their cavity from a top view. This depends on the way the particles lay on the coverslip during sample preparation, for imaging with an inverted microscope. As a result of this, it is possible to see the roughness of the cavity due to residues of dextran that partly diffused into the main templating phase during the UV-photopolymerization.
- The coefficient of variation (CV) is an important parameter for the uniformity of the particles. State the CV value. Specify the particle density, and I would be happy to see the dense packing of single and double emulsion microspheres.
The authors acknowledge the referee for pointing this out and we agree that would confirm the uniform distribution in size of the particles. We stated the coefficient of variation in the manuscript (figure 4a, 5a, S7, S8 and S9). The number of particles (n=50) measured from each sample is mentioned in section 2.5.4. “Microgels Shrinking and Swelling Measurements” and in caption of the respective figures. Bright field images of both microspheres and asymmetric microparticles are added in Figure 6.
- Why are no microscope images shown in the study of swelling and shrinkage? This is one of the most important aspects of this study. An image of swelling and shrinkage with the control should be included in the manuscript.
We agree with the comment of the referee. We included images of the microparticles shrinking and swelling in Figure 6.
- In the previous reports of the same group (Teora et al., Chem. Commun. 2022, 58, 10333, a thermosensitive hydrogel was synthesized based on poly(ethylene glycol) (PEGDA) 25 wt%, PNIPAm 10 wt%, and dextran 20 wt%. I am curious why 30% PEGDA is used in this study? On what basis was it selected and how does it affect yield and mechanical and physical properties?
We acknowledge the referee for their interest in our previous work and for this question. We indeed reported a thermo-responsive system in which the concentration of PEGDA is 25% w/w and PNIPAm 10% w/w. This means that the total polymer w/w percentage in the main phase is 35% w/w, while the concentration of dextran is 20% w/w. In order to obtain an aqueous-two-phase separating system, each polymer solution needs to have values above a certain concentration, otherwise, the two phases would mix. After investigating several factors, the above-mentioned concentrations were chosen to ensure the phase separation of the main composite phase from the dextran, leading to the formation of droplet-in-droplet morphologies that turn into asymmetric shapes after photo-crosslinking. The novelty of this system is that, on top of achieving the phase separation between composite polymer-polymer solution, we successfully found the right concentration of PNIPAm within the system to achieve thermo-responsiveness. In the present work, taking advantage of the knowledge we gained by previous investigation on aqueous-two-phase systems, we decided to use a concentration of 30% w/w PEGDA and 1.5% or 0.75% w/w ALMA, which makes a total polymers concentration in the main phase 31.5% or 30.75% w/w, respectively. This still leads to the generation of asymmetric microgels due to phase separation, with only the anisotropic particles with a higher concentrations of ALMA showing pH-responsiveness. So, in this case, the novelty of our systems is a “never-reported before” composite main polymer phase that shows pH-responsiveness and gives phase separation with a corresponding phase. A higher concentration of ALMA cannot be used due to the increase in solution viscosity, which makes it difficult to inject into the microchannels of the microfluidic device.
By comparing the two microgels, namely PEGDA-PNIPAm: dextran and PEGDA-ALMA: dextran, we can say that the yield in particles formation is the same, as we found the conditions to work during injection for particle generation to give the optimal performance of the microfluidic device. Overall, the mechanical properties of the thermo- and pH-responsive microgels are somewhat comparable in the non-triggered state, but they differ after being exposed to the stimuli. This is in line with the changes in the chemical structure of each stimuli-responsive material in response to its respective triggering stimulus.
- In section 3.7, the maximum drug uptake of 12 µg in 24 hours is determined and a double efficacy is found for anisotropic particles. Can the authors provide the relationship between drug-loaded particles (number of particles) and cell viability? Just wanted to know how the quantification was done. How many particles are needed in the sample to uptake 12 µg in 24 hours?
We thank the referee for the suggestion. We have explained in section 2.7.1. how the loading capacity was calculated, and we added equation 4 for clarity. Unfortunately, it is difficult to quantify the number of particles, therefore we chose to quantify the total mass (µg) of particles used.
Round 2
Reviewer 1 Report
The authors carefully addressed the points raised by the previous reviewers and, from my side, I don't have any additional observations. The manuscript, in my opinion, is ready for publication.
Author Response
We are happy the the reviewer 1 is satisfied with our revision.
Reviewer 2 Report
This manuscript developed a asymmetric PEGDA-ALMA hydrogel with the drug-carrying capacity of 5-FU and the ability to release the drug only at pH 7.4 but not at pH 2.0, which is a promising candidate delivery system with oral and effective targeting of colon cancer. The results are interesting, and might be helpful for the development of oral administration system. However, before the consideration of the publication, improvements are still necessary.
1. The abstract and title need to be revised. More emphasis should be placed on your carriers' potential for oral application in colon cancer rather than pH response. You can refer to this literature, β‐lactoglobulin–pectin Nanoparticle‐based Oral Drug Delivery System for Potential Treatment of Colon Cancer.
2. The cytotoxicity of 5-FU-loaded PEGDA-ALMA hydrogels can be better done with a colon cancer cell line, which was more convincing.
3. In the introduction, there have been many reports on electro-responsive and temperature-responsive hydrogels recently, which can be discussed (Int. J. Biol. Macromol., 2019, 140, 255–264; Acta Biomater., 2018, 72: 55 -69; Bioact. Mater. 2022, 13, 191-199; ACS Nano 2022, 16 , 9859-9870).
4. The title of the vertical axis of Figure 5a should not be expressed in units, it can be changed to a dual-axis graph to distinguish 2R and 2r. Figure S7a and S8a has the same question.
5. Please add a part for explaining how you perform the statistical analysis.
Author Response
We have compiled the response to this reviewer in a response letter.

Reviewer 3 Report
The authors addressed all my concerns. I would recommend the manuscript for publication in Pharmaceutics.
Author Response
We are happy to see the reviewer is satisfied with our revision and response.